# High Incidence of Acute Kidney Injury in Patients Treated with High-Dose Amoxicillin and Cloxacillin Combination Therapy

**DOI:** 10.3390/antibiotics11060770

**Published:** 2022-06-04

**Authors:** Yvon Ruch, Axel Ursenbach, François Danion, Fanny Reisz, Thierry Nai, Baptiste Hoellinger, Yves Hansmann, Nicolas Lefebvre, Jonas Martzloff

**Affiliations:** 1Department of Infectious and Tropical Diseases, Hôpitaux Universitaires de Strasbourg, 67000 Strasbourg, France; axel.ursenbach@chru-strasbourg.fr (A.U.); francois.danion@chru-strasbourg.fr (F.D.); baptiste.hoellinger@chru-strasbourg.fr (B.H.); yves.hansmann@chru-strasbourg.fr (Y.H.); nicolas.lefebvre@chru-strasbourg.fr (N.L.); 2Department of Pharmacy, Hôpitaux Universitaires de Strasbourg, 67000 Strasbourg, France; fanny.reisz@chru-strasbourg.fr (F.R.); thierry.nai@chru-strasbourg.fr (T.N.); 3Department of Nephrology, Hôpitaux Universitaires de Strasbourg, 67000 Strasbourg, France; jonas.martzloff@chru-strasbourg.fr

**Keywords:** acute kidney injury, amoxicillin, cloxacillin, combination therapy, endocarditis

## Abstract

High-dose amoxicillin and cloxacillin combination therapy is recommended for the empiric treatment of selected patients with infective endocarditis despite a low level of evidence. The main objective of this study was to evaluate the renal tolerance of high-dose intravenous amoxicillin and cloxacillin combination. We studied 27 patients treated with amoxicillin and cloxacillin (≥100 mg/kg daily) for at least 48 h. The primary endpoint was the occurrence of acute kidney injury (AKI). The median patient age was 68 ± 8 years, and 16 (59%) were male. The indication for this combination therapy was suspected or confirmed endocarditis with no bacterial identification in 22 (81%) patients. The primary endpoint occurred in 16 (59%) patients after initiating this combination therapy within an average of 4.4 ± 3.6 days. Among them, seven (26%) patients developed severe AKI, including four (15%) patients who required hemodialysis. Other risk factors for AKI were identified in all patients, including injection of iodinated contrast media in 21 (78%), acute heart failure in 18 (67%), cardiac surgery in 11 (41%), and aminoglycoside use in 9 (33%) patients. This study reports an incidence of 59% of AKI after initiating amoxicillin and cloxacillin combination therapy in a population at high renal risk.

## 1. Introduction

Infective endocarditis (IE) remains a therapeutic challenge associated with high mortality rates [1]. Apart from valvular surgery, prompt and appropriate antibiotic therapy is the cornerstone of the treatment. Despite a low level of evidence, the European Society of Cardiology (ESC) Guidelines recommended in 2015 a combination of ampicillin (12 g/day), (flu)cloxacillin or oxacillin (12 g/day), and gentamicin (3 mg/kg/day) as the initial empirical therapy for community-acquired native valve or late prosthetic valve IE in acute severely ill patients prior to pathogen identification [2]. Acute kidney injury (AKI) has been previously reported in patients treated with high-dose amoxicillin [3,4] or cloxacillin [5,6], but data on the tolerability of this combination are scarce. Therefore, the aim of this study was to analyze the characteristics of patients treated with a combination of high-dose intravenous amoxicillin and cloxacillin and to assess the renal tolerance of this combination.

## 2. Results

Thirty-two patients treated with a combination of high-dose intravenous amoxicillin and cloxacillin were identified during the study period. Five patients were excluded because the combination therapy was provided for less than 48 h. The characteristics of the cohort are provided in Table 1. The median age was 68 years (interquartile range [IQR], 56–74), and 16 patients (59%) were male.

The combination therapy was given for a mean duration of 7.1 ± 5.5 days, with seven patients (26%) treated for 10 days or more. The daily dose of amoxicillin or cloxacillin was 12 g or more in 21 (78%) and 24 (89%) patients, respectively. The mean weight-based daily dose for each molecule was 145 ± 49 mg/kg for amoxicillin and 161 ± 44 mg/kg for cloxacillin. Amoxicillin was given by continuous infusion in five (19%) patients and by discontinuous infusion of more than 2 g per administration in five (19%) patients. Cloxacillin was given by continuous infusion in 17 (63%) patients and by discontinuous infusion > of more than 2 g per administration in 6 (22%) patients. Indication for this combination therapy was suspected or confirmed IE with no bacterial identification in 22 (81%) patients. The remaining five (19%) patients were treated for a staphylococcal infection associated with another bacterial identification, mainly enterococci.

Sixteen patients (59%) developed AKI after initiating the combination therapy within an average of 4.4 ± 3.6 days (Table 2). Among them, seven (26%) patients developed Kidney Disease Improving Global Outcomes (KDIGO) stage 3 AKI, including four (15%) patients who required hemodialysis. Involvement of the antibiotic therapy in the occurrence of AKI was considered very likely in two (13%) patients and possible in eight (50%). Other risk factors for AKI were identified in all patients, including the injection of iodinated contrast media in 21 patients (78%), acute heart failure in 18 patients (67%), aminoglycoside use in 9 patients (33%), and cardiac surgery in 11 patients (41%). Among the five patients treated with discontinuous infusion of amoxicillin > 2 g per administration, four developed AKI. Five (19%) patients died, but their death could not be related directly to the antibiotic therapy. All other patients recovered their renal function.

Among the eight patients who developed KDIGO stage 2 or 3 AKI, they were all treated with daily doses of 12 g of amoxicillin and 12 g of cloxacillin. Sepsis (5/8 vs. 1/11; *p* = 0.04) and cardiac surgery (5/8 vs. 1/11; *p* = 0.04) were significantly associated with AKI (Table 3). The main daily dose and weight-based daily dose of amoxicillin was higher in patients with stage 2 or 3 AKI than in those who did not develop AKI, but these differences were not significant (12.0 vs. 9.6 g/day; *p* = 0.07 and 169 vs. 131 mg/kg/day; *p* = 0.09, respectively).

## 3. Discussion

In this retrospective study, we found a high incidence (59%) of AKI in patients treated with high-dose amoxicillin and cloxacillin combination therapy. Moreover, 26% of patients developed KDIGO stage 3 AKI. All patients had other risk factors for AKI, and the majority of these patients were treated for suspected or confirmed endocarditis. Apart from patients who died, all patients recovered their renal function.

Previous studies have found a high renal risk in patients with IE, with approximately a third of patients developing AKI [7,8], especially following cardiac surgery [9]. Our study found a higher rate of renal impairment than previously described in IE. This rate was also higher than the incidence of AKI reported in patients treated with high-dose amoxicillin or cloxacillin alone [6,10]. Petersen et al. reported an incidence of 5.7% of dialysis-requiring AKI in a nationwide cohort of IE [11]. The percentage of dialysis-requiring AKI was three times higher in our study (14.7%). Aminoglycosides use is associated with AKI [9,12], and some authors consider that they are no longer of interest for IE treatment [13]. There were also more diabetic patients among those who developed AKI, although the difference was not significant. This can be explained by a lack of statistical power due to a small sample size. Overall, in this study, the other risk factors for AKI were found to be more important in patients with KDIGO stage 2 or 3 AKI, especially sepsis, cardiac surgery, and aminoglycoside use. We believe that this combination therapy should be used with caution in patients with other risk factors for AKI, including diabetes mellitus.

To our knowledge, the tolerance and the effectiveness of combining these two penicillins has never been studied. Despite this, the 2015 ESC Guidelines have recommended this approach due to its broad spectrum on Gram-positive bacteria commonly involved in IE [2]. The presumed mechanism of AKI with aminopenicillin is indirect tubular toxicity by intratubular crystallization, but acute interstitial nephritis has also been described [3,4,10]. This first mechanism could explain why AKI is associated with a higher daily dose of amoxicillin in this study, even if this difference did not reach statistical significance. Cloxacillin is suspected to be responsible for rare AKI by acute immuno-allergic interstitial nephritis, and this toxicity is not dose-dependent [5,6]. The nephrotoxicity mechanism when combining these two antibiotics remains unknown, but it may result from the toxicity of amoxicillin alone, which is then possibly increased by the addition of cloxacillin. Continuous infusion should be preferred when high doses of amoxicillin are used. Furthermore, discontinuous infusions of more than 2 g per administration should no longer be used, as illustrated by the high incidence of AKI in the patients found in this study (80%).

An alternative to antistaphylococcal penicillin may be a first-generation cephalosporin, which would be effective against methicillin-susceptible staphylococci and would avoid the combination of two penicillins, thus being less nephrotoxic [14]. Nevertheless, the potential lower efficacy of cefazolin in methicillin-susceptible *Staphylococcus aureus* deep-seated infections with high inoculum, such as IE, has been suggested [15]. Daptomycin is an interesting alternative, and data pertaining to its efficacy in endocarditis are increasing [16]. However, there are concerns regarding its potential failure with monotherapy and the emergence of resistance [17,18], which would subsequently encourage the use of a combination therapy. Daptomycin can be combined with gentamicin, and in vitro studies have suggested a synergistic activity with beta-lactams against Gram-positive bacteria [19].

This observational study did not allow us to establish a causal link between the amoxicillin and cloxacillin combination and AKI. However, this real-life study highlights that patients with suspected IE who may benefit from this combination therapy have also multiple comorbidities and numerous risk factors for AKI.

## 4. Materials and Methods

We performed a retrospective study in the Strasbourg University Hospital, which is a 2000-bed tertiary care center located in the northeast of France. We included all patients aged ≥ 18 years, hospitalized in the Strasbourg University Hospital that were treated with high-dose intravenous amoxicillin (≥100 mg/kg daily) and cloxacillin (≥100 mg/kg daily) combination for at least 48 h between January 2016 and December 2020. Patients treated for less than 48 h were excluded. As all prescriptions are carried out by a computer, we screened eligible patients from the prescription software. Data were extracted from medical records.

The primary outcome of this study was the occurrence of AKI among the patients treated with the amoxicillin and cloxacillin combination therapy, according to the KDIGO classification [20]. In addition, the secondary outcomes of this study involved the evaluation of the characteristics of the antibiotic therapies, the time between combination therapy initiation and AKI, other risk factors for AKI, antibiotic imputability in AKI, and all-cause in-hospital mortality rates. Antibiotic imputability in AKI was established by three blinded physicians, including a nephrologist.

This study was approved by the Ethics Committee of the Strasbourg University Hospital (registration number: CE-2021-115) and was registered under ClinicalTrials.gov (NCT05142891). According to the French legislation, we sought the non-opposition of all patients.

The study population was characterized using descriptive statistics. Categorical data were compared using Fischer’s exact test. Continuous data were compared using Student’s *t*-test when applicable (Gaussian distribution of the variable and equal variance) or the Mann–Whitney U test. A *p* value < 0.05 was considered significant.

## 5. Conclusions

This observational study found a high incidence of AKI among patients treated with a high-dose of amoxicillin and cloxacillin. This combination therapy should, therefore, be used with caution in IE, especially since alternative regimens covering the same antibacterial spectrum are available and probably less nephrotoxic in this population with several other risk factors for AKI. Clinical randomized trials are warranted to confirm these results.

## Figures and Tables

**Table 1 antibiotics-11-00770-t001:** Characteristics of patients treated with high-dose amoxicillin and cloxacillin combination.

Characteristics		*n* = 27
Age, median (IQR), years Gender, male		68 (56–74)
		16 (59.3%)
Medical history	Diabetes mellitus	9 (33.3%)
	High risk for infective endocarditis ^1^	7 (25.9%)
	Chronic kidney disease	4 (14.8%)
Weight	Median (IQR), kg	78 (63–89)
	<60 kg	5 (18.5%)
	>100 kg	3 (11.1%)
Duration of combination therapy	Mean ± SD, days	7.1 ± 5.5
	<5 days	12 (44.4%)
	≥10 days	7 (25.9%)
Amoxicillin	Daily dose, mean ± SD, grams	10.6 ± 2.8
	Weight-based daily dose, mean ± SD, mg/kg	145 ± 49
	≥12 g/day	21 (77.8%)
	Continuous infusion	5 (18.5%)
	Discontinuous infusion > 2 g per administration	5 (18.5%)
Cloxacillin	Daily dose, mean ± SD, grams	11.6 ± 1.6
	Weight-based daily dose, mean ± SD, mg/kg	161 ± 44
	≥12 g/day	24 (88.9%)
	Continuous infusion	17 (63.0%)
	Discontinuous infusion > 2 g per administration	6 (22.2%)
Other risk factors for AKI	Iodinated contrast medium	21 (77.8%)
	Acute heart failure with diuretic treatment	18 (66.7%)
	Cardiac surgery with ECC	11 (40.7%)
	Aminoglycoside	9 (33.3%)
	Sepsis	5 (18.5%)
Indication for combination therapy	Infective endocarditis (suspected or confirmed) with no bacterial identification	22 (81.5%)
	Two different bacteria isolated ^2^	5 (18.5%)

Abbreviations: AKI, acute kidney injury; ECC, extracorporeal circulation; IQR, interquartile range; SD, standard deviation. ^1^ Patients with prosthetic cardiac valve or with a previous episode of infective endocarditis. ^2^
*S. aureus* + *E. faecalis* (*n* = 3), *S. lugdunensis* + *E. faecalis* (*n* = 1), *S. epidermidis* + *P. mirabilis* (*n* = 1).

**Table 2 antibiotics-11-00770-t002:** Outcome of patients treated with high-dose amoxicillin and cloxacillin combination.

Characteristics		*n* = 27
Acute kidney injury	KDIGO stage 1	8 (29.6%)
	KDIGO stage 2	1 (3.7%)
	KDIGO stage 3	7 (25.9%)
	Requiring hemodialysis	4 (14.8%)
Time between combination therapy initiation and AKI, mean ± SD, days		4.4 ± 3.6
Imputability of antibiotic therapy in AKI (*n* = 16)	Very likely	2 (12.5%)
	Possible	8 (50.0%)
	Unlikely	6 (37.5%)
Outcome	All-cause in-hospital mortality	5 (18.5%)
	Death directly related to antibiotic therapy	0
	Renal recovery, apart from deaths (*n* = 11)	11 (100.0%)

Abbreviations: AKI, acute kidney injury; KDIGO, Kidney Disease Improving Global Outcomes; SD, standard deviation.

**Table 3 antibiotics-11-00770-t003:** Comparison between patients with KDIGO stage 2 or 3 AKI and patients without AKI.

Characteristics		No AKI (*n* = 11)	KDIGO Stage 2/3 AKI (*n* = 8)	*p*
Age, median (IQR), years Gender, male		68 (58–69)	74 (67–77)	0.17
		4 (36.4%)	5 (62.5%)	0.37
Diabetes mellitus		3 (27.3%)	5 (62.5%)	0.18
Chronic kidney disease		2 (18.2%)	1 (12.5%)	1.00
Weight, median (IQR), kg		84 (59–91)	77 (66–83)	0.90
Duration of combination therapy, mean ± SD, days		7.4 ± 4.1	9.6 ± 8.1	0.45
Amoxicillin	Daily dose, mean ± SD, grams	9.6 ± 3.6	12.0 ± 0	0.07
	Weight-based daily dose, mean ± SD, mg/kg	131 ± 50	169 ± 40	0.09
	≥12 g/day	7 (63.6%)	8 (100.0%)	0.10
	Continuous infusion	3 (27.3%)	2 (25.0%)	1.00
	Discontinuous infusion >2 g per administration	1 (9.1%)	2 (25.0%)	0.55
Cloxacillin	Daily dose, mean ± SD, grams	10.7 ± 2.2	12.0 ± 0	0.13
	Weight-based daily dose, mean ± SD, mg/kg	155 ± 55	169 ± 40	0.54
	≥12 g/day	8 (72.7%)	8 (100.0%)	0.23
	Continuous infusion	8 (72.7%)	4 (50.0%)	0.38
	Discontinuous infusion >2 g per administration	2 (18.2%)	3 (37.5%)	0.60
Other risk factors for AKI	Iodinated contrast medium	7 (63.6%)	7 (87.5%)	0.34
	Acute heart failure with diuretic treatment	6 (54.5%)	6 (75.0%)	0.63
	Cardiac surgery with ECC	1 (9.1%)	5 (62.5%)	0.04
	Aminoglycoside	2 (18.2%)	5 (62.5%)	0.07
	Sepsis	1 (9.1%)	5 (62.5%)	0.04
Infective endocarditis (suspected or confirmed)		8 (72.7%)	7 (87.5%)	0.60

Abbreviations: AKI, acute kidney injury; ECC, extracorporeal circulation; IQR, interquartile range; KDIGO, Kidney Disease Improving Global Outcomes; SD, standard deviation.

## Data Availability

All data is contained within the article.

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
