# Peer review of "High Incidence of Acute Kidney Injury in Patients Treated with High-Dose Amoxicillin and Cloxacillin Combination Therapy"

_antibiotics, 2022, doi:10.3390/antibiotics11060770_

Round 1

Reviewer 1 Report

Dear Editor,

MDPI-Antibiotics

I have evaluated the manuscript (Antibiotics-1737666) titled “High Incidence of Acute Kidney Injury in Patients Treated with High-Dose Amoxicillin and Cloxacillin Combination Therapy” by Ruch and coworkers, and the author has demonstrated that 

acute kidney injury (AKI) is associated with the high-dose amoxicillin and cloxacillin combination therapy used for the treatment of selected patients with infective endocarditis. All standard methods were used for this research. I found this article interesting for the readers and follow the scope of the journal Antibiotics. I don’t have any major comments as this article is well written, however, the author could have taken more sample size. 

I would recommend the article could be published in Antibiotics after minor corrections. 

The author needs to address the following comments/corrections.

  1. The author needs to mention the details of Strasbourg University Hospital, a tertiary care center, and its location.
  2. All the results could be in tabular form for better understanding.
  3. The author could include the following references

(a) Crochette R, Ravaiau C, Perez L, Coindre JP, Piccoli GB, Blanchi S. Incidence and Risk Factors for Acute Kidney Injury during the Treatment of Methicillin-Sensitive Staphylococcus aureus Infections with Cloxacillin Based Antibiotic Regimens: A French Retrospective Study. J Clin Med. 2021;10(12):2603. Published 2021 Jun 12. doi:10.3390/jcm10122603

(b) Lavergne A, Vigneau C, Polard E, Triquet L, Rioux-Leclercq N, Tattevin P, Golbin L. Acute kidney injury during treatment with high-dose cloxacillin: a report of 23 cases and literature review. Int J Antimicrob Agents. 2018 Sep;52(3):344-349. doi: 10.1016/j.ijantimicag.2018.04.007. Epub 2018 Apr 14. PMID: 29665445.

Author Response

We thank reviewer 1 for his good review and his constructive comments.

  1. As suggested, we added more details on our institution: “Strasbourg University Hospital, a 2000-beds tertiary care center located in the north-east of France” (line 153-154).
  2. All the results are exposed in the 3 tables.
  3. As suggested, these two intersting references have been included [5,6] and have enriched the content of the manuscript.

Reviewer 2 Report

The authors should compare other antibiotic combinations for endocarditis in the article - checking the rate of kidney damage and percentage of recovery from this endpoint.

Author Response

We thank reviewer 2 for his comment. Data on acute kidney injury in infective endocarditis depending on the antibiotic therapy are scarce. Globally, the rate of AKI in our cohort was higher than the incidence reported in IE patients. We enriched the discussion with other data (lines 99 to 105), especially regarding AKI under amoxicillin or cloxacillin alone. We found no data on the percentage of recovery of AKI in IE patients.

Reviewer 3 Report

This is an interesting retrospective study on the potential nephrotoxic effect of high dose amoxicillin and cloxacillin for empiric treatment of adult patients with suspected infective endocarditis. My main concern is the potential confounding effect of diabetes (33% of your patients) and chronic renal disease (14.8% of your patients) on the incidence of AKI with combination therapy.

Table 3 reveals a 62.5% incidence of AKI in your diabetic patients vs. 27.3% in patient without diabetes. The p value is not significant, but this likely represents a Type II error given your small sample size. I would suggest addressing this in your discussion. Perhaps clinicians should be especially careful with respect to combination use in this population.

I did not see that the CKI patients were addressed. Did you compare the incidence of AKI with combination therapy in this cohort vs. patients without CKI? Alternatively, should these patient be excluded? I also could not find a list of exclusion criteria in your Materials and Methods section.

Finally, with respect to the diabetes population and a potential Type II error, did you perform a power calculation, and if so what did you consider to be the smallest clinically important difference in the incidence of AKI with combination therapy?

Author Response

We would like to thank reviewer 3 for these positive and constructive comments.

First, it is absolutely right that diabetes was more prevalent in patients with AKI, as reported in table 3. Even if the difference with patients without AKI was not significant, we enriched our discussion on this specific point (lines 100-116). We agree that this could be due to a lack of stastical power due to the small sample size, and we mentionned that limit (line 112).

There were 4 patients with chronic kidney disease among all patients. That could be seen on table 1. These patients were not excluded from the study. We used the same definition (KDIGO) to screen AKI in these patients. Among these 4 patients, only one developped stage 2 KDIGO AKI, another developped stage 1 AKI and the 2 remaining did not develop any AKI. These data were added on Table 3.

The only exclusion criteria was combination therapy for less than 48h. We precised that point on the Materials and Methods section: “Patients treated for less than 48h were excluded.” (line 157-158)

We did not perform a power calculation but included all patients meeting the inclusion criteria during the study period. We agree that the sample size is, unfortunately, not important.

Round 2

Reviewer 3 Report

Thank you for addressing my concerns. I agree with your explanations.